# L138ins Variant of the *CFTR* Gene in Russian Infertile Men

**DOI:** 10.3390/genes14071407

**Published:** 2023-07-07

**Authors:** Vyacheslav Chernykh, Tatyana Sorokina, Anna Sedova, Maria Shtaut, Olga Solovova, Ekaterina Marnat, Tagui Adyan, Tatyana Beskorovaynaya, Anna Stepanova, Olga Shchagina, Aleksandr Polyakov

**Affiliations:** 1Research Centre for Medical Genetics, 115522 Moscow, Russia; reprolab@med-gen.ru (T.S.); luoravetlanka@gmail.com (A.S.); shtaut@yandex.ru (M.S.); olga_pilyaeva@list.ru (O.S.); adyan@dnalab.ru (T.A.); t-kovalevskaya@yandex.ru (T.B.); cany@yandex.ru (A.S.); schagina@med-gen.ru (O.S.); polyakov@med-gen.ru (A.P.); 2Pirogov Russian National Research Medical University of the Ministry of Healthcare of the Russian Federation, 117997 Moscow, Russia; marnat_eg@rsmu.ru

**Keywords:** azoospermia, male infertility, gene variant, *CFTR* gene, Cystic Fibrosis, CBAVD

## Abstract

(1) Introduction: Pathogenic variants in the *CFTR* (Cystic Fibrosis Transmembrane conductance Regulator, OMIM: 602421) gene cause Cystic Fibrosis (CF, OMIM: 219700) and CF-related disorders (CF-RD), often accompanied by obstructive azoospermia due to congenital bilateral aplasia of vas deferens (CBAVD, OMIM: 277180) in male patients. The L138ins (c.413_415dup; p. (Leu138dup)) is a mild variant in the *CFTR* gene that is relatively common among CF-patients in Slavic populations. The frequency of this variant in Russian infertile men has not been sufficiently studied; (2) Materials and Methods: The sample consisted of 6033 Russian infertile men. The patients were tested for 22 common in Russian populations pathogenic variants of the *CFTR* gene and the IVS9Tn-polymorphic locus of the intron 9. Molecular-genetic studies were performed using amplified fragment length polymorphism (AFLP-PCR), multiplex ligation-dependent probe amplification (MLPA), and nested PCR (for analysis of the IVS9Tn-polymorphic locus); (3) Results: Pathogenic variants in the *CFTR* were detected in 3.9% of patients. The most frequent variants were F508del and CFTRdele2.3(21kb), accounted for 61.0% and 7.1% of detected variants, respectively. The L138ins variant was detected in 17 (0.28%) individuals: one of them was homozygous, 10 patients were heterozygous, and 6 patients were compound-heterozygous (F508del/L138ins, *n* = 4; L138ins/N1303K, *n* = 1; L138ins/5T, *n* = 1). Two pathogenic CF-causing variants in the *CFTR* gene were detected in 8 patients, including 7 compound heterozygous (F508del/L138ins, *n* = 4; F508del/N1303K, *n* = 1; 2184insA/E92K, *n* = 1; 3849+10kbC>T/E92K, *n* = 1) and one homozygous (L138ins/L138ins). The L138ins variant was found in 7 out of 16 (43.75%) chromosomes in six of these patients. The most common pathogenic variant, F508del, was identified in five out of them, in 5 of 16 (31.25%) chromosomes. The allele frequency (AF) of the L138ins variant in the sample has been found to be 0.0014.; (4) Conclusions: The L138ins variant of the *CFTR* gene is the third most common variant after F508del and CFTRdele2.3(kb) among Russian infertile men.

## 1. Introduction

Pathogenic variants in the *CFTR* gene (Cystic Fibrosis Transmembrane conductance Regulator, OMIM: 602421) are known to cause one of the most common autosomal recessive monogenic diseases, Cystic Fibrosis (CF, OMIM: 219700).

Besides CF, pathogenic *CFTR* gene variants are also associated with the development of CF-related diseases (CF-RD), in particular CBAVD syndrome, recurrent acute or chronic pancreatitis, chronic sinusitis or rhinosinusitis, disseminated bronchiectasis, allergic bronchopulmonary aspergillosis and asthma [1]. In addition, pathogenic and certain polymorphic variants in the *CFTR* are frequently observed genetic factors contributing to male infertility [2,3,4].

Commonly, the frequency of *CFTR* gene variants has been found to be higher in infertile male patients compared to the general population, including Russian men with impaired fertility without diagnosed CF [5,6]. This is mainly due to Congenital Bilateral Aplasia of Vas Deferens (CBAVD, OMIM: 277180), which is characterized by obstructive azoospermia [2,7]. This excretory form of male infertility can be associated with CF or can be isolated (CBAVD syndrome) [2,3,4]. On average, about 80% of patients with CBAVD syndrome have pathogenic variants and/or the 5T (IVS9T5) allele in the *CFTR* gene; certain ethnic groups show a different frequency and spectrum of these variants [2,3,4,8,9]. The meta-analysis of the IVS9-5T variant in different populations showed that it was one of the risk factors for impaired spermatogenesis, and this was more pronounced in non-European men [10].

The *CFTR* gene encodes the CFTR protein, which acts as a chloride channel and is predominantly expressed in epithelial cells [1]. More than 2000 pathogenic variants in the *CFTR* have been identified and documented in the database of pathogenic variants in the CFTR (CFTR2 database; https://cftr2.org/, accessed on 4 April 2023). The frequency and spectrum of the *CFTR* gene variants vary significantly in different regions of the world. In a number of countries, including the Russian Federation, the most frequently observed pathogenic variant in the *CFTR* is c.1521_1523delCTT (F508del; p.(Phe508del)) [11,12,13,14]. In addition, Russian populations frequently carry pathogenic variants, including CFTRdele2,3(21kb), E92K, 1677delTA, 3849+10kbC>T, 2143delT, W1282X, N1303K, and G542X, with an allele frequency (AF) greater than 1% in patients, diagnosed with CF [12,13,14,15].

The L138ins variant (c.413_415dupTAC, rs397508679, p.(Leu138dup)) is a mild pathogenic variant that is classified as a class IV-V mutation in the *CFTR* gene based on its mechanism of pathogenicity. This variant is one of the most common in CF patients of Eastern European origin, especially in the Volga-Ural region [11,16]. The allele frequency (AF) of the L138ins variant in Russian CF patients was 0.0153, as documented in the Register of Cystic Fibrosis Russian Patients (2018); however, this frequency varied from 0 to 0.028 depending on the region within the Russian Federation [11]. Therefore, the L138ins variant can be classified as one of the most frequently observed pathogenic variants in the *CFTR* gene in Russian CF-patients. The L138ins variant of the *CFTR* gene was detected previously in infertile men with CBAVD syndrome [6,8,17], however, its frequency in patients with impaired fertility, especially in Russian men with reproductive problems, has not been adequately studied.

The aim of this study is to determine the frequency of the L138ins variant in the *CFTR* gene and the *CFTR* genotypes of carriers in Russian infertile male patients.

## 2. Materials and Methods

The sample consisted of 6033 unrelated Russian men aged 18–60 years old from infertile couples, having male infertility factor associated with abnormal sperm characteristics (pathozoospermia). This large cohort of infertile and subfertile men was studied at the Research Centre for Medical Genetics (RCMG) during a longitudinal complex genetic and andrological examination (karyotyping by standard chromosome analysis, screening for Y chromosome microdeletions and pathogenic *CFTR* gene variants, and analysis of CAG repeats of exon 1 of the androgen receptor, AR gene, semen analysis) carried out between 2002 and 2022. The majority of examined individuals had male infertility associated with idiopathic azoospermia or oligozoospermia. Patients diagnosed with CF prior to the study were not included.

Written informed consent was obtained from all participants before the study was conducted. The study was approved by the Bioethics Committee of the RCMG.

A standard semen analysis was performed according to the WHO laboratory manual for the examination and processing of human semen (2010) [18].

Blood was obtained by venipuncture and collected in disposable plastic test tubes containing a preservative solution (0.5 M EDTA) at a ratio of 1:10 (preservative: blood). Genomic DNA was isolated using Wizard Genomic Purification Kit from Promega (Madison, WI, USA).

The patients were tested for 22 common in Russian populations pathogenic variants of the *CFTR* gene (c.54-5940_273+10250del21kb (CFTRdele2,3), c.262_263delTT (394delTT), c.274G>A (E92K), c.413_415dupTAC (L138ins), c.472_473insA (604insA), c.489+1G>T (621+1G>T), c.1000C>T (R334W), c.1521_1523delCTT (F508del), c.1545_1546delTA (1677delTA), c.1624G>T (G542X), c.2012delT (2143delT), c.2051_2052delAAinsG (2183AA>G), c.2052dupA (2184insA), c.3140-26A>G (3272-26A>G), c.3587C>G (S1196X), c.3691delT (3821delT), c.3718-2477C>T (c.3717+12191C>T; 3849+10kbC>T), c.3816_3817delGT (3944delGT), c.3846G>A (W1282X), c.3883delA (4015delA), c.3890_3891insT (4022insT), c.3909C>G (N1303K)). Also, the IVS9Tn-polymorphic locus of the intron 9 of the *CFTR* gene was analyzed.

Molecular-genetic studies were performed using amplified fragment length polymorphism (AFLP-PCR), multiplex ligation-dependent probe amplification (MLPA), and nested PCR (for analysis of the IVS9Tn-polymorphic locus).

PCR was performed using Taq polymerase on a DNA-technology MS2 thermal cycler. The reaction mixture contained genomic DNA, dNTPs, PCR buffer, thermophilic DNA polymerase, and mineral oil. The PCR consisted of an initial denaturation at 95 °C for 2 min, followed by 35 cycles at 94 °C for 45 s, 65 °C for 45 s, and 72 °C for 45 s, with final completion at 72 °C for 7 min. Results were evaluated by electrophoresis on a 7% polyacrylamide gel stained with ethidium bromide (Figure 1, Table 1) [5,18].

The methodology used for the molecular genetic analysis of the *CFTR* gene is described in detail in our previous studies [5,18].

## 3. Results

Pathogenic variants were detected in 300 (2.49%) of the analyzed alleles of the *CFTR* gene in 292 (4.84%) of 6033 infertile Russian men. Twenty-two different mutations were detected in the studied cohort. Among the carriers of pathogenic CF-causing genetic variants, 284 individuals were heterozygotes, 7 patients were compound heterozygotes and one patient was homozygote. The most common pathogenic variant was the F508del found in 183 (3.03%) patients, including 179 heterozygous and 4 compound-heterozygous (F508del/L138ins) individuals. The allele frequency (AF) of this genetic variant was 0.01517, accounting for 61% of all identified CF-causing variants of the *CFTR* gene in our cohort (Table 2).

Other commonly identified variants in the *CFTR* gene were c.54-5940_273+10250del21kb (CFTRdele2,3), c.413_415dupTAC (L138ins), and c.3846G>A (W1282X). Three these variants account for 18% of the total number of detected variants. Eight genetic variants, including c.1545_1546delTA (1677delTA), c.3718-2477C>T (3849+10kbC>T), c.274G>A (E92K), c.2012delT (2143delT), c.1624 G>T (G542X), c.2052dupA (2184insA), c.3909C>G (N1303K), and c.1000C>T (R334W) were identified in 51 patients (Table 2). In total, these variants accounted for 17.5% of all detected *CFTR* gene variants, with the frequency of each individual mutation being greater than 1%. The other detected variants were only found in one or two alleles of the *CFTR* gene.

We found two pathogenic CF-causing variants in the *CFTR* gene in eight patients, seven of which were compound heterozygous (F508del/L138ins, *n* = 4; F508del/N1303K, *n* = 1; 2184insA/E92K, *n* = 1; 3849+10kbC>T/E92K, *n* = 1) and one was homozygous (L138ins/L138ins). The L138ins variant (c.413_415dup or c.411_412insCTA, p.(Leu138dup)) was found in 7 out of 16 (43.75%) chromosomes in six of these patients. The most common pathogenic variant, F508del, was identified in five out of them, in 5 of 16 (31.25%) chromosomes. The L138ins variant was detected in 17 (0.28%) patients, including 11 heterozygous, one homozygous, and five compound heterozygous with other common CF-causing variants in the *CFTR* gene (F508del, *n* = 4; N1303K, *n* = 1).

One patient was found to be compound heterozygous for the 5T allele, which is often associated with CBAVD syndrome. Other carriers of the L138ins variant had 7T/7T (*n* = 9) and 7T/9T (*n* = 7) genotypes for the IVS9Tn polymorphic locus of the *CFTR* gene (Table 3). Thus, the allele frequency (AF) of L138ins in examined cohort of Russian infertile men was 0.15%, making it the third most common *CFTR* gene variant after F508del and CFTRdele2,3(21kb). No heterozygous carriers of the L138ins variant, including one patient with diagnosed with CBAVD syndrome in combination with the 5T (IVS9T5) allele, had symptoms of CF (Table 3).

The prevalence of genotypes with two CF-causing genetic variants in our sample of infertile Russian men was 1 per 872. The patients with two CF-causing variants in the CFTR gene, including those with the L138ins variant in their genotype, have obstructive azoospermia due to CBAVD syndrome. In addition, clinical signs of previously undiagnosed CF are characterized by chronic pancreatitis and/or respiratory disorders such as chronic sinusitis, chronic bronchitis or non-allergic bronchial asthma.

## 4. Discussion

The L138ins variant (c.413_415dupTAC, p.(Leu138dup)) is a duplication of three CTA-nucleotides in exon 4 of the *CFTR* gene. This type of genetic variant belongs to small insertions/deletions that do not change the reading frame, also known as in-frame insertions/deletions [16,19]. This nucleotide sequence variant results in a leucine duplication (CTA codon) at position 138, which adds an amino acid residue to CFTR protein molecule. This position is located in the second motif of the CFTR protein penetrating the membrane, the first membrane–bound domain (MSD1) involved in the formation of the pore of the chlorine channel, which violates the properties of the conductivity of the chlorine channel (IV-V classes of pathogenic effect on the CFTR protein) [19].

The L138ins variant in the *CFTR* gene was first discovered in 1996 in a study by Dörk T. et al. in 1 of 106 patients with CBAVD syndrome [8]. The patient had the L138ins/5T genotype, preserved pancreatic function, no lung involvement, and normal sweat chloride (less than 60 mmol/L). Currently, the L138ins variant (c.413_415dupTAC, p.(Leu138dup)) is defined as pathogenic, causing CF-causing variant of the *CFTR* gene with mild clinical signs [19,20].

According to the Registry of Russian Patients with Cystic Fibrosis [11], the L138ins variant is one of the common genetic variants of the *CFTR* gene in Russian patients with CF (Table 4).

In general, the allele frequency (AF) of this genetic variant in Russian CF patients is 0.0153, but it obviously depends on age, e.g., in children (under 18 years) it is 1.43%, in adults—1.83% [11]. The slightly higher frequency of this variant in *CFTR* in adult patients is probably due to the milder phenotype of CF in its carriers. Thus, compared to F508del/F508del homozygotes, F508del/L138ins compound heterozygotes have milder clinical manifestations of CF. They have lower sweat chloride levels and a later age at diagnosis of CF, they more often have a higher FEV1 index, their nutritional status is better due to preserved exocrine pancreatic function. F508del/L138ins compound heterozygotes do not have meconium ileus, CF-associated diabetes mellitus, severe liver damage and chronic Pseudomonas aeruginosa (*Pseudomonas aeruginosa*) infection is much less common than those who are homozygous for the most common pathogenic *CFTR* gene variant, F508del [19,20].

In our recent study of a cohort of adult Russian men diagnosed with CF, we found four patients who were compound heterozygotes for the L138ins variant (F508del/L138ins, *n* = 3; 2184insA/L138ins, *n* = 1) [7]. All patients developed pancreas sufficient CF (PS-CF) and obstructive azoospermia due to congenital bilateral aplasia of vas deferens. In the present study, all 7 compound heterozygotes for the L138ins variant also had obstructive azoospermia/CBAVD (Table 3). In addition, clinical signs of previously undiagnosed CF are characterized by chronic pancreatitis and/or damage to the respiratory system such as chronic sinusitis, chronic bronchitis, or non-allergic bronchial asthma. Evidently, men with the L138ins variant may be fertile or infertile depending on the *CFTR* genotype and the presence or absence of CF/CBAVD, sperm parameters and other factors affecting male fertility. According to the results of the present study, infertile patients with this genetic variant in homozygous and compound heterozygous L138ins variant with another CF-causing mutation or 5T allele of the *CFTR* gene) have congenital aplasia of the vas deferens (obstructive azoospermia) as a result of CF or CBAVD syndrome. Infertile male patients heterozygous for the pathogenic *CFTR* variant may be non-azoospermic with variable sperm parameters. A majority of patients with CBAVD/obstructive azoospermia can have offspring using by IVF/ICSI procedure with testicular spermatozoa [2,3,4].

According the results of genetic evaluation of Russian infertile men, the L138ins variant accounted for 6% of all detected variants in the *CFTR* gene, ranking third in frequency after F508del and CFTRdele2,3. According to the CF Registry (2020), the L138ins variant shares 9/10th place with the N1303K variant in the general group of Russian CF patients, with a frequency of 1.53%. Variability in the frequency of the L138ins variant was observed in different regions of the Russian Federation. Specifically, in patients with CF in the Northwestern Federal District its frequency was 1.1% (ranked 10th), in St. Petersburg—1.1% (ranked 12th), in the Volga Federal District—2.2% (ranked 5th), in Moscow—2.0% (ranked 9th), and in the Ural Federal District—2.8% (ranked 3rd) [11]. The frequency of the L138ins variant in the *CFTR* gene, according to the CFTR2 database [https://cftr2.org/ accessed 4 April 2023], is 0.00014. According to the RUSeq Browser database [http://ruseq.ru/ accessed 3 March 2023], the allele frequency (AF) of the L138ins variant among individuals from the European part of Russia is 0.0004. The AF of this variant among non-Finnish Europeans in the EXAC, gnomAD Exome, and gnomAD Genome databases is 0%. Population studies have revealed differences in the frequency of heterozygous carrier status of the L138ins variant in the *CFTR* gene in different regions of Russia. The presence of the L138ins variant was found in residents of the Republic of Tatarstan (AF of the variant was 0.00283) [15], and in the Vologda region (AF—0.0016) [21,22]. In these regions, the L138ins variant was the second most common variant in the *CFTR* gene after the F508del.

Recently, Solovyova et al. investigate pathogenic and polymorphic variants in the *CFTR* gene in a sample of 2146 infertile men. The L138ins variant was detected in 7 (0.33%) individuals (0.163% of the examined chromosomes), including four patients with azoospermia in the heterozygous state, one patient with CBAVD in the compound heterozygous state (genotype F508del/L138ins, 9T/7T), and in two infertile men (one patient with azoospermia and one patient with asthenozoospermia) in the combination with the 5T (IVS9T5) allele of *CFTR* gene [6]. The proportion of the L138ins variant among all detected CF-causing variants of the *CFTR* gene was 10% (7 out of 70). Previously, in a smaller sample of infertile Russian men (*n* = 963), we found the L138ins variant in two individuals [5]. This sample of Russian infertile men is more than 6 times larger. The increase in sample size allowed us to detect more patients with *CFTR* gene mutations, including more patients with the L138ins variant (17 individuals compared to two). It should be noted that the L138ins variant was later included in the panel of common *CFTR* mutations in Russian CF patients, as well as in the panel of *CFTR* variants in Russian infertile men. This is because these *CFTR* mutation panels have changed over time. It is likely that the increase in sample size has led to a change in the frequency of individual mutations. In addition, it cannot be excluded that the frequency and spectrum of *CFTR* variants could be influenced by sampling characteristics. It is noteworthy that 6 out of 8 patients with two pathogenic CF-causing *CFTR* mutations carried the L138ins variant. This study has confirmed previous data also showing that infertile men with non-severe *CFTR* genotypes may develop mild, atypical forms of CF or CF-related disorders, may have undiagnosed Cystic Fibrosis, also as CF and CBAVD are partially overlapping disorders [5].

In the present study, the allele frequency (AF) of the L138ins variant in the studied sample and the AF were close (0.28% and 0.00149, respectively). It should be noted that the sample of patients studied by Solovyova et al. consisted of men examined in Siberia (Krasnoyarsk), whereas our sample consisted mainly of men living in the European part of the Russian Federation, mainly from the central region of Russia, and specifically from Moscow city and the Moscow region. This may have had some influence on the differences in these frequencies. In addition, the frequency and spectrum of pathogenic variants of the *CFTR* gene could be influenced by differences in sample size and patient selection criteria. The total frequency of the L138ins variant in our sample and in the sample of patients studied by Solovyova et al. was found in 24 (0.3%) out of 8177 patients, and its AF was 0.00153, which is close to the data obtained from a population survey of residents of the Vologda region [21].

## 5. Conclusions

According to our data, the L138ins (c.413_415dup; p.(Leu138dup)) variant is one of the most frequent pathogenic variants in the *CFTR* gene in Russian infertile men. Men carrying two pathogenic *CFTR* gene variants, non-severe *CFTR* genotypes, particulary the L138ins variant in combination with another CF-causing variant in cis position, may have undiagnosed (before genetic examination) CF. This possibility should be considered in the evaluation and management of infertile men, especially the patients with obstructive azoospermia/CBAVD or other signs of Cystic Fibrosis.

## Figures and Tables

**Figure 1 genes-14-01407-f001:**
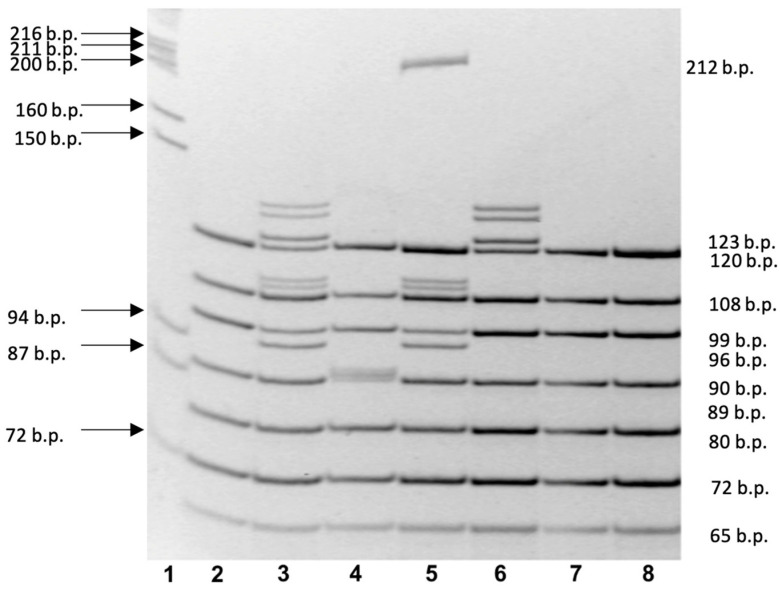
Polyacrylamide gel electrophoresis with results of multiplex PCR analysis of the *CFTR* gene. Line 1—Lambda phage DNA restricted by Pst1 endonuclease (λ/Pst1); lines 2, 7, and 8—normal *CFTR* alleles; Line 3—heterozygous c.1521_1523delCTT (F508del) variant; heterozygous c.413_415dupTAC (L138ins) variant; Line 4—heterozygous c.2052dupA (2184insA) variant; Line 5—heterozygous c.1521_1523delCTT (F508del) and c.54-5940_273+10250del21kb (CFTRdele2,3) variants; Line 6—heterozygous c.413_415dupTAC (L138ins) variant.

**Table 1 genes-14-01407-t001:** *CFTR* gene variants analyzed by amplified fragment length polymorphism (AFLP-PCR) and the length of normal DNA fragment and abnormal DNA fragment in the *CFTR* gene variant, base pairs (b.p.).

*CFTR* Gene Variant *	Length of Normal DNA Fragment (b.p.)	Length of Abnormal DNA Fragment in the *CFTR* Gene Variant (b.p.)
c.54-5940_273+10250del21kb (CFTRdele2,3) *	No fragment	212
c.413_415dupTAC (L138ins)	120	123
c.3816_3817delGT (3944delGT)	108	106
c.1521_1523delCTT (F508del) c.1545_1546delTA (1677delTA)	99	96 97
c.2012delT (2143delT), c.2051_2052delAAinsG (2183AA>G) (c.2052dupA (2184insA))	89	88 90
c.262_263delTT (394delTT)	80	78
c.3691delT (3821delT)	72	71
c.472_473insA (604insA)	65	66

* Variants are given according to traditional nomenclature.

**Table 2 genes-14-01407-t002:** Pathogenic *CFTR* gene variants detected in the cohort of 6033 infertile Russian men.

*CFTR* Gene Variant *	Number of Alleles, Carrying This Variant, *n*	Allele Frequency (AF)	The Percentage of Variant among All Detected *CFTR* Gene Variants, %
F508del	183 ^1^	0.01517	61.00
CFTRdele2.3(21kb)	20	0.00166	6.67
L138ins	18 ^2^	0.00149	6.00
W1282X	16	0.00133	5.33
1677delTA	8	0.00066	2.67
3849+10kbC>T	8 ^3^	0.00066	2.67
E92K	8 ^4^	0.00066	2.67
2143delT	7	0.00058	2.33
G542X	6	0.00049	2.00
2184insA	6 ^5^	0.00049	2.00
N1303K	4 ^6^	0.00033	1.33
R334W	4	0.00033	1.33
3821delT	2	0.00017	0.67
2183AA>G (Lys684Ser)	2	0.00017	0.67
3242-26A>G	1	0.00008	0.33
4015delA	1	0.00008	0.33
604insA	1	0.00008	0.33
621+1G>T	1	0.00008	0.33
G551D	1	0.00008	0.33
G85E	1	0.00008	0.33
dup 7-11	1	0.00008	0.33
S1196X	1	0.00008	0.33

* Variants are given according to traditional nomenclature. ^1^ including 179 heterozygous and 4 compound-heterozygous (F508del/L138ins). ^2^ including 11 heterozygous, 5 compound-heterozygous (F508del/L138ins, *n* = 4; L138ins/N1303K, *n* = 1) and one homozygous (L138ins/L138ins). ^3^ including 7 heterozygous and one compound-heterozygous (3849+10kbC>T/E92K). ^4^ including 6 heterozygous and two compound-heterozygous (3849+10kbC>T/E92K, *n* = 1; 2184insA/E92K, *n* = 1). ^5^ including 5 heterozygous and one compound-heterozygous (2184insA/E92K). ^6^ including 3 heterozygous and one compound-heterozygous (L138ins/N1303K).

**Table 3 genes-14-01407-t003:** *CFTR* genotypes and clinical data in 17 patients with the L138ins variant.

*CFTR* Genotype *	Number of Patients, *n*	CBAVD, Obstructive Azoospermia	Extragenital Signs of CF **
L138ins/N, 5T/7T	1	+	-
L138ins/N, 7T/7T	8	-	-
L138ins/N, 7T/9T	2	-	-
L138ins/L138ins, 7T/7T	1	+	+
L138ins/N1303K, 7T/9T	1	+	+
F508del/L138ins, 9T/7T	4	+	+

* *CFTR* gene variants are given according to traditional nomenclature. N—normal allele. ** The presence of CF symptoms affecting other organs or systems besides the reproductive system (the respiratory and digestive systems, namely the lungs, pancreas, intestines, etc.).

**Table 4 genes-14-01407-t004:** Common pathogenic *CFTR* gene variants detected in Russian CF patients.

*CFTR* Gene Variant *	Coding DNA Name	Protein Name	rsID	Allele Frequency (AF) **
F508del	c.1521_1523delCTT	p.(Phe508del)	rs113993960	0.5261
CFTRdele2,3	c.54-5940_273+10250 del21kb	p.(Ser18Argfs*16)	not found	0.0615
E92K	c.274G>A	p.(Glu92Lys)	rs121908751	0.0325
1677delTA	c.1545_1546delTA	p.(Tyr515*)	rs121908776	0.0212
3849+10kbC>T	c.3718-2477C>T	No protein name	rs75039782	0.0211
2143delT	c.2012delT	p.(Leu671*)	rs121908812	0.0202
2184insA	c.2052_2053insA*(c.2052dupA)	p.(Gln685Thrfs*4)	rs121908786	0.0193
W1282X	c.3846G>A	p.(Trp1282*)	rs77010898	0.0173
L138ins	c.413_415dupTAC	p.(Leu138dup)	rs397508679	0.0153
N1303K	c.3909C>G	p.(Asn1303Lys)	rs80034486	0.0153
G542X	c.1624G>T	p.(Gly542*)	rs113993959	0.0146
394delTT	c.262_263delTT	p.(Leu88Ilefs*22)	rs121908769	0.0085
R334W	c.1000C>T	p.(Arg334Trp)	rs121909011	0.0074
*S466X	c.1397C>G	p.(Ser466*)	rs121908805	0.0059
W1282R	c.3844T>C	p.(Trp1282Arg)	rs397508616	0.0055
3821delT	c.3691delT	p.(Ser1231Profs*4)	rs121908783	0.0051
S1196X	c.3587C>G	p.(Ser1196*)	rs121908763	0.0045
1367del5	c.1240_1244delCAAAA (c.1243_1247delAACAA)	p.(Asn415*)	rs397508184	0.0040
2789+5G>A	c.2657+5G>A	No protein name	rs80224560	0.0038
R1066C	c.3196C>T	p.(Arg1066Cys)	rs78194216	0.0038
3272-16T>A	c.3140-16T>A	No protein name	rs767232138	0.0033
W1310X	c.3929G>A	p.(Trp1310*)	not found	0.0032
3944delGT	c.3816_3817delGT	p.(Ser1273Leufs*28)	rs397508612	0.0029
712-1G>T	c.580-1G>T	No protein name	rs121908793	0.0022
621+1G>T	c.489+1G>T	No protein name	rs78756941	0.0020
R553X	c.1657C>T	p.(Arg553*)	rs74597325	0.0020
4015delA	c.3883delA	p.(Ile1295Phefs*33)	rs397508630	0.0017
L1335P	c.4004T>C	p.(Leu1335Pro)	rs397508658	0.0017
R785X	c.2353C>T	p.(Arg785*)	rs374946172	0.0017
R1162X	c.3484C>T	p.(Arg1162*)	rs74767530	0.0016
1898+1G>C	c.1766+1G>A	No protein name	rs121908748	0.0014
CFTRdup7-11 (6b-10*)	c.(743+1_744-1)_(1584+1_1585-1)dup	No protein name	not found	0.0014
1898+1G>A	c.1766+1G>C	No protein name	rs121908748	0.0013
R347P	c.1040G>C	p.Arg347Pro	rs77932196	0.0013
3849G>A	c.3717G>A	No protein name	rs144781064	0.0012
G85E	c.254G>A	p.Gly85Glu	rs75961395	0.0012
S1159F	c.3476C>T	p.(Ser1159Phe)	rs397508573	0.0012
3667ins4	c.3535_3536insTCAA (c.3532_3535dupTCAA)	p.(Thr1179Ilefs*17)	rs387906378	0.0012

* Variants are given according to traditional nomenclature. ** Allele frequency of the nucleotide sequence of the *CFTR* gene in Russia (genetic variants with a frequency of more than 0.10% are presented; adopted from [11]).

## Data Availability

Not applicable.

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
