# Peer review of "L138ins Variant of the CFTR Gene in Russian Infertile Men"

_genes, 2023, doi:10.3390/genes14071407_

Round 1

Reviewer 1 Report

   In this manuscript, Chernykh VB et al studied 6033 Russian infertile men. Using AFLP-PCR and MLPA, they found the L138ins variant in the CFTR is the third most common variant after F508del and CFTRdele2.3 (kb) among Russian infertile men.

 Major points:

1. How long have you been collecting the samples?

2. What is the clinical outcome of the L138ins mutation? Can they have offspring through assisted reproduction?

3. Page 1, line 37, please confirm the sentence “without diagnosed CF”.

4. In the author’s previous work, W1282X is the third most common variant after F508del and CFTRdele2.3(kb) among Russian infertile men. While with the sample size increases, L138ins become the third. So you can only say that the L138ins variant in the CFTR is the third most common variant depending on the present research.

5. The main problem with the article is the lack of useful context. The authors do not explain why their new findings are of interest in the field of CFTR or male infertility, considering the authors reported similar results in 2010.6. The authors should indicate the size of bands with markers in Fig1.

No comments.

Author Response

We would like to thank Reviewer 1 for your valuable feedback. Your comments have greatly improved our manuscript. We appreciate your cooperation and hope to work with you in the future.

Point 1:  How long have you been collecting the samples?

Response 1: The patients have been examined in the 22 years’ period (2000-2022). This information was added in the text.

Point 2: What is the clinical outcome of the L138ins mutation? Can they have offspring through assisted reproduction?

Response 2: Men with L138ins variant may be fertile or infertile depending on CFTR genotype and presence or absence of CF/CBAVD, sperm parameters and other factors affecting male fertility. According to the results of the present study, infertile patients with this genetic variant in homozygous and compound heterozygous L138ins variant with other CF-causing mutation or 5T allele of the CFTR gene have congenital aplasia of the vas deferens (obstructive azoospermia) resulting from CF or CBAVD syndrome. Infertile male patients with heterozygous L138ins variant can be non-azoospermic with different sperm parameters. A majority of patients with CBAVD/obstructive azoospermia can have offspring using IVF/ICSI procedure with testicular spermatozoa.

Point 3: Page 1, line 37, please confirm the sentence “without diagnosed CF”

Response 3:  This sentence was corrected. “Commonly, the frequency of CFTR gene variants has been found to be higher in infertile male patients compared to the general population, including Russian men with impaired fertility without diagnosed CF”.

Point 4: In the author’s previous work, W1282X is the third most common variant after F508del and CFTRdele2.3(kb) among Russian infertile men. While with the sample size increases, L138ins become the third. So you can only say that the L138ins variant in the CFTR is the third most common variant depending on the present research.

Response 4: The Reviewer correctly noted that in the previous study, the W1282X variant was the third most frequent, but in the present study, the L138ins variant was more frequent. It should be noted that the L138ins variant was later included in the panel of common CFTR mutations in Russian CF patients, as well as in the panel of CFTR variants for Russian infertile men. Because these CFTR mutation panels have modifying over time. Probably, the increase in the sample size has led to a change in the frequency of individual mutations. In addition, it cannot be excluded that the frequency and spectrum of CFTR variants could be affected by sampling characteristics. So, Russian patients is not russian patients, because of the multiethnic population of Russia.

Point 5: The main problem with the article is the lack of useful context. The authors do not explain why their new findings are of interest in the field of CFTR or male infertility, considering the authors reported similar results in 2010.

Response 5: We are not completely agreeing with this comment, but it is useful for text editing. Actually, we had evaluated CFTR gene variants in a cohort of Russian infertile men, but it was much smaller. Apparently, it is allowed to revealed more patients with CF-causing mutations, particularly, individuals with two pathogenic CFTR gene variants. A most interesting finding of presented study is that L138ins variant is relatively frequent in Russian infertile men, and a most of them with two CF-causing CFTR gene mutations, carrying namely this genetic variant. These patients had azoospermia resulted from Cystic Fibrosis and/or CBAVD. It was added in ‘Discussion’ section. We hope, that our results will be useful to other researchers investigating the CFTR gene and male infertility.

Point 6: The authors should indicate the size of bands with markers in Fig1.

Response 6: Figure 1 has been corrected. 

Reviewer 2 Report

The authors use classical molecular techniques to assay CFTR variants in the western Russain male population. It is notable that the author utilized classical molecular techniques (gel) to assay CFTR variants, namely L138ins in the western Russian male population. The paper generated new discoveries by focusing on a previously less studied population. 

However, there are concerns regarding the high level of self-citation which contribute to possible predatory classification, which has recently been a topic of discussion for MDPI journals. Read more here https://predatoryreports.org/news/f/mdpi-self-citation-problem

https://www.mdpi.com/about/announcements/2979. In total, at least 6/21 (28%) citations are self-citation.

Specifically, the methodology section alone cites 13 articles to say the method used is also performed elsewhere. Among the 13 papers, at least 5 (38%) papers were authored by the same group, also at least 5 of the cited articles are in Russian with no English translation.

This possibly raises questions about the reliability of such techniques. Is AFLP-PCR still the industry standard technique in the world? 

I recommend the author to perform SNP arrays or CFTR region sequencing, such as this https://pubmed.ncbi.nlm.nih.gov/30296588/.  If the author intends to continue using gel techniques to assess CF variants, more information could be considered. 

  1. It is fair to say that the method is used in other groups, but the current paper does not even list the primer sets used. More information should be provided regarding the primer used and enzyme used.

  2. Only one gel is shown for 7 samples from 6033 human data points. It is not possible for a reviewer to judge the validity or reliability of such data without seeing the data. I suggest the author upload the genotyping data in a cloud service such as https://figshare.com/. which support upto 20GB. Assuming a gel picture around 1MB could represent 10 data points, 600MB is well below the 20GB threshold. If that is too much, only sharing the 300 data point with CF variants is also possible.

Overall, to enhance the credibility of the paper, it is recommended that the author address the concerns regarding self-citation and provide additional details and data to support their findings.

One minor note, I suggest the author to further define “extragenital regions” in Table 2, I interpret as other epithelia of the body besides Vas Deferens / Epididymis regions (lung, intestine, sweat gland, pancreas). Not sure if that is authors' intention.

Author Response

We thank reviewer 2 for his valuable comments. We have tried to take all comments into account

Reviewer 3 Report

Dear authors, congratulations on your paper. Even though I think it's interesting, I believe it requires major revision. 

Introduction, line 33 - please, list some examples of CF related disorders.

Introduction, lines 48-50 - the F508del is the most common mutated variant among Caucasians in general, do not limit your paper to the Russian population only. Also, your references for this fact are only from Russian sources, you could include other articles, analyzing the prevalence in Europe, not only in Russia. 

Materials and Methods, lines 67-68 - please, be more specific about the sperm impairments of the included subjects, pathozoospermia is too general. Also, please, give more details about your patients - did they have normal karyotype, were they tested for Y microdeletions, did they have normal levels of different sex hormones, etc. You did not go into any details about your patients - you could have provided, for example, mean age, details about their sperm samples, etc. 

Materials and Methods, lines 86-89 - I believe that this is a too detailed description of your protocol, there is no need to list the amount of microliters. 

Figure 1 - it should be improved, line 1 is curved and not clear.

Results - lines 107-109, 124-125, 128-130 - your findings are not presented in an understandable way. It is not clear what is the number of the heterozygotes, of the homozygotes, of the compound heterozygotes, you could have listed this in Table 1, not just the total number of the alleles. CF is an autosomal recessive disorder, it is not a dominant one, that is why this is important information. 

Table 2 - you did not write what does N stand for. maybe for normal, but it should have been written in the legend. 

Results, lines 139 - 143 - sounds more like Discussion, why did you put it in the Results section?

Discussion, lines 169 - 173 - were patients with CF included in your study? This is not clear. If this is a study for the role of L138ins variant in the CFTR gene in the etiology of infertility, then CF patients maybe should be excluded from the study? Again, please, define your study criteria. 

Discussion - you could include that CF is associated also with non-obstructive azoospermia - CFTR gene variants as a reason for impaired spermatogenesis: a pilot study and a Meta-analysis of published data: Human Fertility: Vol 25, No 4 (tandfonline.com)

All the best!

Author Response

We thank reviewer 3 for his valuable comments. We have tried to take all comments into account

Round 2

Reviewer 2 Report

The authors made notable attempts in improving the manuscripts.

However, it is still problematics in two areas in the issues I documented in first round review.

1. High levels of self-citation, exhibit A, a standard Methodology description of PCR and gel running techniques perhaps does not need all 14 citations in line 159-160. The authors intend to justify this by claim of small field working on Russian CF. It is not surprising to have a small group working on the specific population of CF variants, given it is a genetic disease. The citations does not have to be limited to such a scope focusing on Russian, it can be of any population or even any diseases. The purpose of that citation is to support the techniques and its application to genotype a region of a genome. The PCR then gel running to genotype CF are used back in the 90s. One or two of those citations would be sufficient to support the point. 

2. Not sufficient data to backup the claims in the manuscripts of allele frequencies. Claim 8177 data points, data only shown for 7. An acceptable alternative can be 300 Pathogenic variants data. The authors' excuse of not have the summary or database of all relevant data point further put the reliability of these datapoint in question.

Additional details can be found in a more detailed written first review.

The author is on the right path, I believe with proper documentation and reduced level of self-citation, this manuscript could be a top quality piece.

Author Response

https://drive.google.com/drive/folders/1uFULA0_H1xNqd26MZsTYSShR70DfJPXx?usp=share_link

Here is a link to the table with the pathogenic variants detected in our sample and an archive with photos of the electrophoresis.

Reviewer 3 Report

Dear authors, I appreciate you made the manuscript corrections I had suggested. I have no further recommendations. All the best! 

Author Response

Author's response to Reviewer 3.
